# Impact of Baking Powder and Leavening Acids on Batter and Pound Cake Properties

**DOI:** 10.3390/foods12050946

**Published:** 2023-02-23

**Authors:** Eugenia Ayebea Asamoah, Alain Le-Bail, Anthony Oge, Delphine Queveau, Olivier Rouaud, Patricia Le-Bail

**Affiliations:** 1Oniris, Nantes Université, CNRS, GEPEA, UMR 6144, F-44000 Nantes, France; 2INRAE, UR 1268, Biopolymères Interactions Assemblages, BP 71627, CEDEX, 3F-44316 Nantes, France; 3USC 1498 INRAE-TRANSFORM Department and GEPEA UMR CNRS 6144, Rue de la Géraudière, 44316 Nantes, France

**Keywords:** cake, specific volume, porosity, baking powder, SAPP—sodium acid pyrophosphate, carbon dioxide

## Abstract

In most soft wheat products such as cakes, baking powder (BP) plays an important role in achieving the desired product volume through batter aeration by the release of CO_2_ during baking. However, the optimization of a blend of constituents in BP is minimally documented, especially the selection of acids, which is often supported by the suppliers based on their experience. The objective of this study was to evaluate the impact of two sodium acid pyrophosphate leavening acids (SAPP10 and SAPP40) at different levels in BP on final pound cake properties. A central composite design of the response surface methodology (RSM) was used to design the blend ratio of SAPP with different amounts of BP to investigate some selected cake parameters such as specific volume and conformation. Results showed that increasing the BP level significantly increased the batter specific volume and porosity but dropped as BP approached maximum (4.52%). The batter pH was influenced by SAPP type; SAPP40 presented a relatively sufficient neutralization of the leaving system as compared to SAPP10. Furthermore, lower BP levels resulted in cakes with large air cells, which presented a non-homogeneous crumb grain. This study therefore highlights the need to identify the optimum amount of BP to attain the desired product qualities.

## 1. Introduction

One of the most desired qualities in bakery products, typically in cakes, is having a soft texture obtained by a high specific volume. Indeed, the porous and aerated structure yields a soft texture and a high level of eating quality [1]. To achieve this, it is imperative to have a proper balance of leavening acid and base in the cake formulation. Therefore, chemical leavening plays an important role by providing acidification to release carbon dioxide (CO_2_), buffering to give optimum pH within the baked product and modifying the elastic and viscosity properties of doughs and batters [2].

The bases are the CO_2_ sources in chemical leavening systems, and among them sodium bicarbonate (NaHCO_3_), also known as baking soda, is the most popular within the bakery industry [3,4]. However, using this alone as a leavening agent will result in an inadequate release of CO_2_ as well as an increase in the alkalinity as it dissolves within the cake matrix [4]. This may be detrimental to the product by affecting the flavor and color, usually, a soapy flavor and yellow color in the resultant product [5]. To avoid this, a leavening acid must be added to enable the release of the remaining CO_2_ by lowering the pH to produce a balanced neutralization reaction.

In determining the appropriate quantity of leavening acid to add, the neutralization value (NV) is considered. NV indicates “the number of parts by weight of sodium bicarbonate that can be neutralized (cause to release all available carbon dioxide gas) by 100 parts by weight of the leavening acid” [4].

With several acids available to be used for chemical leavening in cakes, there is a need to appropriately select a leavening acid(s) to adequately facilitate the release of CO_2_ during the setting of the product [4,6]. Owing to this, leavening acids are classified by the time and rate at which they become soluble and able to react with NaHCO_3_ to produce CO_2_ gas. They can therefore be either fast-acting leavening acids which solubilize rapidly and react mostly during batter mixing (e.g., monosodium phosphate), or slow-acting which solubilize more slowly and react to produce CO_2_ mostly at baking temperatures [2,4,7].

Among these leavening acids, sodium acid pyrophosphate (SAPP), a slow-acting leavening acid, is one of the most commercially used. It is usually associated with a number that corresponds to an estimated percentage of CO_2_ gas generated during mixing [4,6]. This presents several grades of SAPP such as SAPP 10, SAPP 28 and SAPP 40, which indicates that ~10%, 28% and 40% of the available CO_2_ will be released during batter mixing (termed as cool action), while the remaining 90%, 72% and 60%, respectively, will be released during baking (thus, hot action) [4].

Despite the vast amount of information available about how leavening systems function [2,4,7,8,9], few studies have been done on the use and optimization of blend constituents in baking powder (BP), especially the selection of acids, which is often supported by the suppliers based on their experience.

A study by Book and Brill [6] explored the effect of chemical leavening systems on the properties of a yellow cake formula with low, medium and high levels of NaHCO_3_ using five different types of leavening acids. Godefroidt et al. [10], on the other hand, compared the functionality of organic and inorganic leavening acids in cream cake systems, whereas Cepeda et al. [11] and Christaki et al. [12] also evaluated the functionality of combinations of different inorganic leavening acids in wheat flour tortillas and cakes. Most of these studies done so far only involved the use of completely different kinds of leavening acids but not the combinational use of acids within the same kind at different progressive levels in cake matrices.

This study therefore investigates the use of a blend of two leavening acids, specifically SAPP10 and SAPP40 with sodium bicarbonate as the leavening base, altogether known as BP. Different levels of BP were established on one side, and within the leavening acid proportion on the other side, different amounts of the SAPP’s were fixed using the central composite design of the response surface methodology (RSM). These leavening systems were incorporated into a pound cake recipe. Pound cake was of interest because it is one of the highly consumed cakes in France. The use of a chemical leavening agent or BP addresses quality issues in terms of volume and cake structure. Nonetheless, the recommendations about dosage are often based on the know-how of the retail companies manufacturing the BP mixes. This paper intends to propose a more scientific approach to investigate the impact of the total amount of BP and of the type of acidic component on the final cake structure. The results are to be used in a complementary ongoing project aimed at the inflow baking of pound cake using ohmic heating.

## 2. Materials and Methods

### 2.1. Ingredients

Wheat flour type 45 (15.1% water content, 11.5% protein, 1.0% fat, 68.1% starch and 0.5% ash on a wet basis) was supplied by Minoterie Giraudineau, and the pasteurized whole liquid eggs (77.5% water content, 12.1%, protein, 10.2% fat, 0.8% minerals and 0.8% carbohydrates on a wet basis) were purchased from Transgourmet (France). Sucrose (caster sugar) was obtained from Béghin-Say (France), sunflower oil from Transgourmet (France) and refined salt from Cerebros Esco (France). Baking powder (BP), which comprised sodium bicarbonate (NaHCO_3_), was sourced from Louis Francois (France) and both leavening acids (SAPP 10 and SAPP 40—sodium acid pyrophosphate) from Budenheim (Germany).

### 2.2. Experimental Design for Batter and Cake Formulation

The batter formulation used was adapted from Khodeir et al. [13]. However, in this study, BP (leavening acid (SAPP 10 and SAPP 40) + leavening base (NaHCO_3_)) was included (Table 1). The amount of BP added was based on the proportion of 100g of wheat flour (WF). Hence, the calculated BP amount was made up of 41.9% for the base part (NaHCO_3_) and 58.1% for the acid part (SAPP 10 and SAPP40) in order to respect the neutralization value, NV = 72.

For a better representation of the leavening system used with respective amounts of SAPPs and NaHCO_3_, a general formula was generated:**X**%BP = acid (**a**SAPP_10_ + **b**SAPP_40_) + base (NaHCO_3_)(1)
where **X** is the percentage of total BP in the formulation, **a** is the amount of SAPP 10 within the leavening acid, and likewise, **b** is the amount of SAPP 40 within the leavening acid component. This means that **a** + **b** must always be equal to 58.1% of BP. Hence, where SAPP10 increases, SAPP40 decreases and vice versa to achieve the total acid required. On the other hand, as the base component only comprised NaHCO_3_, its proportion was always fixed (41.9%).

With this formula established, we had two experimental factors:Level of total baking powder per formulation.Level of SAPP 10 within the acid component with its complementing amount of SAPP 40.

Owing to this, the maximum and minimum levels of BP chosen with the help of some preliminary testing trials were 4.52% and 0.98%, respectively, based on 100 g of WF in the recipe as explained previously. For the second experimental factor, the maximum and minimum levels selected for SAPP 10 within the leavening acid component were 100% and 0%, which are the mirror image of SAPP 40.

Using the central composite 2^2^ + star design of the response surface methodology (RSM), ten different batter and cake formulations (Table 2) were generated by the Statgraphics Centurion 19 software (Statgraphics Technologies, Inc, The Plains, VA, USA). The design was run as a single block with 2 center points (thus, formulation 6 and 7 as shown in Table 2) and the order of experiments were completely randomized (see Appendix A).

#### 2.2.1. Batter Preparation

The batter was prepared using the KitchenAid equipment (St Joseph Model KSM90, Whirlpool Corporation, MI, USA). First, sugar was added to the wet ingredients (whole egg and sunflower oil), then mixed at speed 6 using the KitchenAid. This was followed by the addition of the remaining dry ingredients (flour, salt and baking powder) at speed 1, after which the speed was increased to 8 to obtain the final batter with a total time of 6 min (Figure 1).

Each formulation was prepared in triplicate, three productions of batter (2000 g each), making 30 batter productions in total. For each batter production, 500 g was poured into three baking pans (each 18 cm × 10 cm × 8 cm) for baking. This made a total of 90 cakes for all the 10 different batter formulations.

#### 2.2.2. Pound Cake Baking

The pound cakes were baked at 180 °C for 45 min using an instrumented conventional oven three cakes at a time. The oven was connected to a CO_2_ measuring device which measured the amount of CO_2_ liberated during baking.

### 2.3. Characterization of Batter and Pound Cakes

#### 2.3.1. Specific Volume Measurements

Immediately after preparing the batters with their respective formulations, the specific volume was measured using 60 mL open mouth syringes whose masses and volumes had been predetermined with water as adapted from Krause et al. [14]. The batters were slowly drawn into the syringes to avoid the destruction of air bubbles. After determining their weights, the specific volumes were calculated by ratio of the volume of syringes to the mass of batter in cm^3^/g.

For the cakes, they were cooled for 2–3 h after baking, then stored and sealed in plastic bags overnight at room temperature (day 0). On day 1, the weights of the cakes were determined, and the volumes were measured with a Volcalc laser-based scanner (Perten Instruments, TexVol, BVM-L370LC, Paris, France). Likewise, the specific volumes were calculated by the ratio of cake volume to the mass (cm^3^/g).

#### 2.3.2. Porosity Measurements

With the calculation of specific volume, the inverse gives the apparent batter density ($\rho_{app},$ g/cm^3^) which was used to calculate the porosity of the batter (air volume fraction—Φa) against the true density of the batter (thus without air bubbles). The true batter density ($\rho_{true}$) was therefore determined by the sum of the densities of each ingredient by their volume fraction in the recipe. Hence, porosity in percentage was calculated by the following equation adapted from Assad Bustillos et al. [15].
(2)$$\Phi a_{\;}=(1-\frac{\rho_{app}}{\rho_{true}})\;\times \;100\%$$

Similarly, the porosity of cakes was determined using the equation above. Apparent cake density ($\rho_{app}$), was obtained from the inverse of their specific volumes, whereas its true density ($\rho_{true}$) was measured using a helium pycnometer Accupyc 1330 (Micromeritics Instruments Corporation, Merignac, France).

#### 2.3.3. Batter pH

The pH value of the batter was measured with a solid-state pH electrode (VWRcollection-662-0084, Avantor Incorporated, PA, USA) using about 10–20 g of the prepared batter with their respective formulations.

#### 2.3.4. Conformation of Pound Cakes

The measurement of cake conformation was simply based on the height after baking and cooling. Specifically, the total cake height at the center and the base height (thus excluding the top expansion outside the baking pan or which does not take the shape of the baking pan) using a metric ruler.

All analyses were performed in triplicate.

### 2.4. Statistical Analysis

Analysis of variance (ANOVA) was conducted using the Statgraphics Centurion 19 software (Statgraphics Technologies, Inc., The Plains, VA, USA). The Fisher’s least significant difference (LSD) test was used to identify significant differences (*p* < 0.05) in the batter and cake properties with respective levels of BP and SAPPs.

## 3. Results and Discussion

From the ten formulations generated by the central composite design of the response surface methodology (Table 2), it can be seen that there were mainly five different levels of BP and SAPP10/SAPP40.

### 3.1. Effect of SAPP and BP Levels on Batter Properties

In the batter making process, air is physically introduced while mixing, and when an agent of expansion such as baking powder is added, its cold action produces one part of CO_2_ which solubilizes within the aqueous and lipidic batter phases and another part which expands the existing air bubbles [6]. This aerates the batter system which helps decrease batter density and impacts upon the batter specific volume and porosity, which may further have an influence on the final cake product [15].

The decomposition of sodium bicarbonate allows the release of CO_2_, being 10% and 40% when mixing (cold action) for SAPP 10 and SAPP 40, respectively, whereas the remaining is released during the baking process.

From the response surfaces in Figure 2a,b, it can be observed that at low levels of BP, the batter specific volume and porosity are highest but drop significantly when SAPP40 becomes predominant (thus, 0% of SAPP10) at elevated levels of BP. Conversely, at predominant levels of SAPP10, both low and high levels of BP present a more stable and uniform batter specific volume and porosity.
**These observations can be explained by two hypotheses:**
The CO_2_ released during the decomposition reaction of NaHCO_3_ solubilizes in the water and fat present in the batter.Afterwards, the excess of non-solubilized CO_2_ disperses into the air bubbles trapped in the batter during mixing/beating, leading to an increase in the size of the bubbles and thus favoring coalescence and the loss of “gas” (air + CO_2_) from the surface.

Owing to this, SAPP10 does not release enough CO_2_ for the second hypothesis to occur (as only 10% of available CO_2_ is released in the batter), hence corresponding to a lower production of CO_2_ gas leading to relatively smaller-sized air bubbles with adequate spaces between them. As a result, the solubilized air cells are more stable to prevent bubble coalescence, hence maintaining a more stable batter specific volume and porosity at 100% SAPP10 even as the level of BP increases (Figure 2a,b).

On the other hand, for SAPP40, the quantity of non-solubilized CO_2_ is very significant (as 40% of available CO_2_ is released in the batter), hence a relatively higher production of CO_2_ gas especially at high levels of BP. As a result, the air bubbles become strongly inflated by the CO_2_ gas, creating fewer interfacial spaces between the air cells and provoking a high tendency of bubble coalescence which leads to the bursting and escape of the “gas” (air + CO_2_) bubbles from the surface of the batter, being visible to the naked eye. This therefore results in a reduced batter specific volume at high levels of BP with maximum quantity of SAPP40. Batter porosity also follows a similar trend.

In terms of pH, it can be observed in Figure 2c that although it does not evolve with the increase of BP in cake batter, it is however very different between the use of SAPP 10 and SAPP 40. The batters prepared with 0% SAPP10 (thus, 100% SAPP40) had the lowest pH (6.61), whereas those prepared with 100% SAPP10 (thus, 0% SAPP40) had the highest pH (7.33). This reveals that the amount of CO_2_ released and solubilized when using SAPP10 is not sufficient to observe a decrease in the pH. On the contrary, the amount of sodium bicarbonate which did not react during the cold action is relatively higher (~90%) which leads to an increase in its pH, unlike SAPP40 whereby the amount of solubilized CO_2_ leads to a significant acidification of the batter.

### 3.2. Influence of Levels of BP and SAPP Type on Pound Cake Properties

#### 3.2.1. Specific Volume, Porosity and CO_2_ Liberated

Comparing the batter specific volume and porosity values to that of cake from Figure 2 to Figure 3, it can be observed that both properties increased over double fold from batter to cake (Figure 3a,b). This could be attributed to the major quantity of CO_2_ gas produced in the oven during baking (thus, the remaining 90% and 60% of the available CO_2_ gas from SAPP10 and 40, respectively).

In producing a cake with a porous structure and high volume, batter expansion is the initial step that occurs. It involves the previously incorporated air bubbles during mixing, which now act as a nuclei for CO_2_ produced from BP to diffuse into, causing them to expand and grow in size [4,16]. This is followed by the loss of moisture as water evaporates and volume rises to reach a maximum until the cake structure is set by heat (starch gelatinization and protein denaturation), and the air pockets are captured inside the cake [16]. Several studies have shown that early production of CO_2_ gas before structure setting will lead to the loss of the inflated gas cells, which may result in cakes with low volumes and/or coarse crumbs [4,17,18].

Contrarily, when the leavening system reacts too late, the CO_2_ gas produced after the product has set may no longer be able to expand the air cells, but rather induce cracks and blowouts in the crumb due to the high pressure. Consequently, the cake may collapse by the rupturing of the air cells and weakening of the cake structure [2,4,7,18].

In this study, significant differences were found in all cake properties as BP level increased (Table 3). Cakes formulated with 0.98% BP had the lowest specific volume, but at 1.5% and 2.75%, specific volume increased progressively. However, when BP level further increased to 4.52%, the specific volume of cake decreased as shown in Figure 3a. A similar trend was identified for the porosity of cakes in their response surfaces as shown in Figure 3b.

This could be attributed to higher production of CO_2_ during baking, which significantly enlarges the air bubbles coupled with the heat and mass transfer that is accompanied by starch gelatinization and protein denaturation as explained previously. In other words, cake specific volume and porosity were influenced by the amount of expanded CO_2_ gas bubble that can be contained during baking as the structure sets. Therefore, at the maximum level of BP, the amount of CO_2_ generated may be too much to be contained in the cake matrix considering the phase transition from fluid batter to a solid cake through starch gelatinization and protein denaturation which interplayed with the movement of water and food molecules causing a significant loss of air + CO_2_ gas bubbles. This led to the decline of cake specific volume and porosity at high levels of BP.

Furthermore, the measured amounts of CO_2_ liberated within the oven during baking shows a positive correlation with increasing BP levels as expected as shown in Figure 3c.

Concerning the leavening acid type (SAPP10 and 40), its effect appeared to be more constant with no noticeable impact on the response surfaces of cake specific volume, porosity and CO_2_; however, when analyzed statistically, significant differences were detected unlike that found in the batters (Table 3).

#### 3.2.2. Influence on Cake Conformation

In terms of cake conformation, no significant differences were found for total cake height as SAPP10 increases although the response surface in Figure 4a appears to tilt a bit lower at elevated levels of SAPP10. However, when only the base height (Figure 4b) is considered, the elevated level of SAPP10 has more influence by increasing cake base height as BP level increases compared to SAPP40 at maximum level.

It is worth noting that lower amounts of BP (F9—1.5% and F10—0.98%, with predominantly SAPP40) resulted in cakes with large air cells, which presented a non-homogeneous crumb grain (Figure 5). Crumb grain is defined as the exposed cell structure of crumb when a leavened baked product is sliced. It consists of a solid phase apparent in the cell wall structure and a fluid phase made up of air [19].

According to Hayman et al. [20], crumb grain can be categorized as open if it is composed of intermediate to large cells (thus, non-homogeneous), and closed if it is composed of small uniform gas cells (thus, homogeneous). With this said, F9 and F10 cakes can be considered to have an open crumb grain. This could be due to the fact that at low levels of BP, relatively lower amounts of CO_2_ are generated during baking; hence, the CO_2_ could possibly be fully diffused into the incorporated air bubbles and expand them to their largest capacity during baking which makes them highly susceptible to coalesce. The coalesced gas cells lead to fixated large gas cells as the cake structure sets when starch gelatinized, and protein denatures. As a result, the large air pockets formed present a less attractive crumb grain in these cakes (F8–F10 of Figure 5).

In addition, particularly for the F10 cakes, the final products appeared to have a top crumb region, which was not fully cooked (Figure 5, circled in red) even with the same oven baking conditions as all the other formulations which were fully cooked. This characteristic could certainly be due to the very little CO_2_ released which limits the full expansion of the cake, hence leaving it with a relatively compacted crumb.

A companion study by Hayman et al. [21] on bread crumb grain suggests that crust formation restricts the expansion of gas cells in the unbaked portion, hence creating a pressure too great for the cell walls to withstand. This further leads to the rupture and coalescence of the gas cells resulting in an open crumb grain as previously discussed.

## 4. Conclusions

The use of an experimental design was successfully applied to investigate the effect of a blend of two leavening acids (SAPP10 and 40) using increasing amounts of baking powder. This helped to better understand and explore the influence of leavening acid type and BP amounts on batter properties and cake quality.

In summary, the levels of BP significantly influenced batter specific volume and porosity, while on the other hand, the leavening acid type significantly influenced batter pH. In terms of cake properties, the different levels of both baking powder and SAPPs caused extremely significant differences independently. However, when the cross effects of both constituents (BP × SAPPs) were considered, both batter and cake properties were less affected.

Additionally, this study has proved that BP plays an essential role in the final appearance of cakes, especially the crumb grain. It is therefore expedient to be within an optimum range of BP level to have uniform diffusion of CO_2_ in air cells to ensure no or minimum coalescence in order to produce a globally aerated and porous product. This could possibly avoid the occurrence of uncooked crumb regions resulting from compacted crumb with limited and non-uniformed expansion mostly occurring when relatively low levels of BP are used.

Therefore, in the case of this study, cakes at the 2.75% BP level appear to stand out with the highest cake specific volume and porosity as well as the greatest height. This makes it an interesting formulation to consider, with the most attractive cake qualities falling within the midlevel of BP and together with 100% SAPP10. Hence, with the goal of having a deeper understanding of the influence of baking powder and SAPP type in a pound cake system, together with the effect of the baking process on the overall product quality, the 2.75% BP level could be considered for future studies.

## Figures and Tables

**Figure 1 foods-12-00946-f001:**
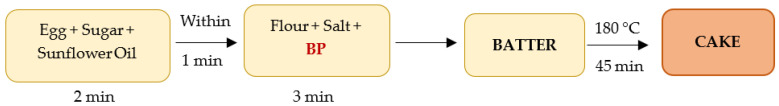
Batter and pound cake making protocol.

**Figure 2 foods-12-00946-f002:**
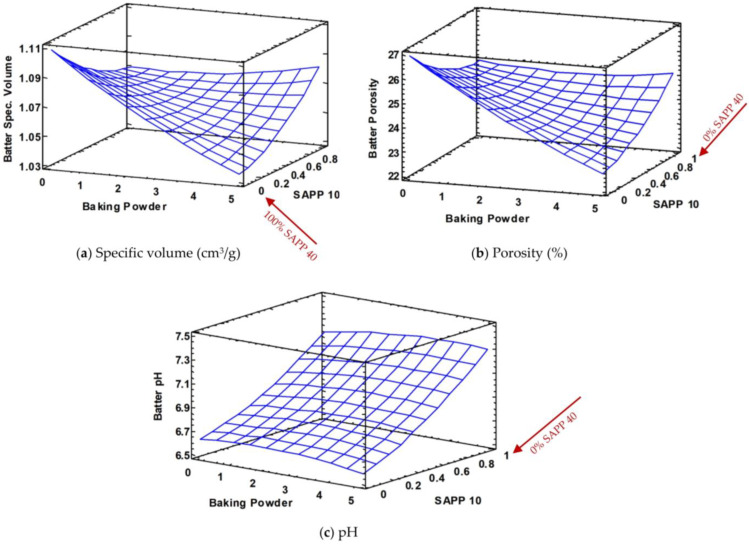
Response surfaces of batter specific volume (**a**), porosity (**b**) and pH (**c**) with respect to baking powder (BP) and sodium acid pyrophosphate (SAPP) levels (red arrow indicates SAPP 40 is the inverse of SAPP 10).

**Figure 3 foods-12-00946-f003:**
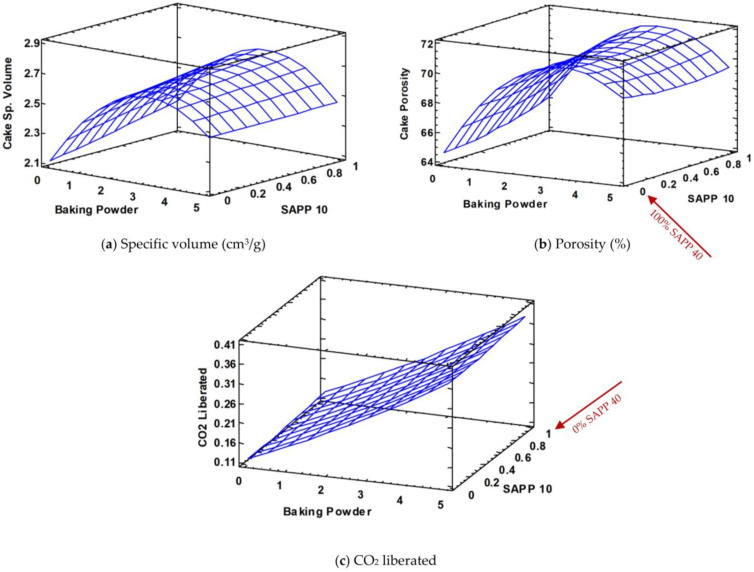
Response surface of pound cake specific volume (**a**), porosity (**b**) and CO_2_ liberated (**c**) during baking with respect to baking powder and SAPP levels (red arrow indicates SAPP 40 is the inverse of SAPP 10).

**Figure 4 foods-12-00946-f004:**
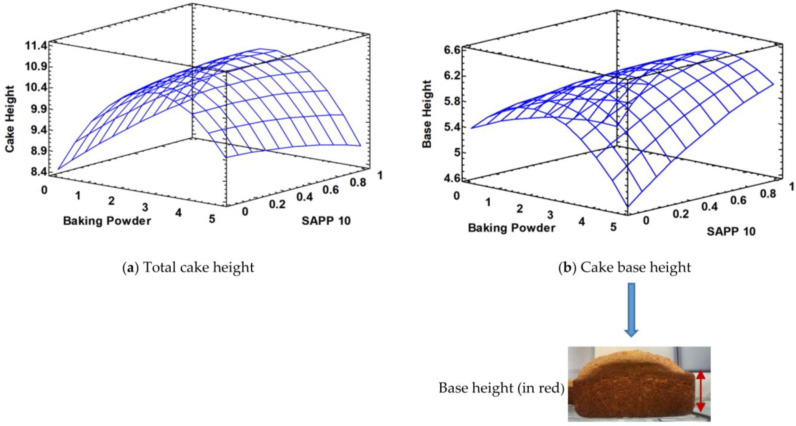
Response surfaces of total (**a**) and pound cake base (**b**) heights with respect to baking powder (BP) and SAPP levels (SAPP 40 is the inverse of SAPP 10).

**Figure 5 foods-12-00946-f005:**
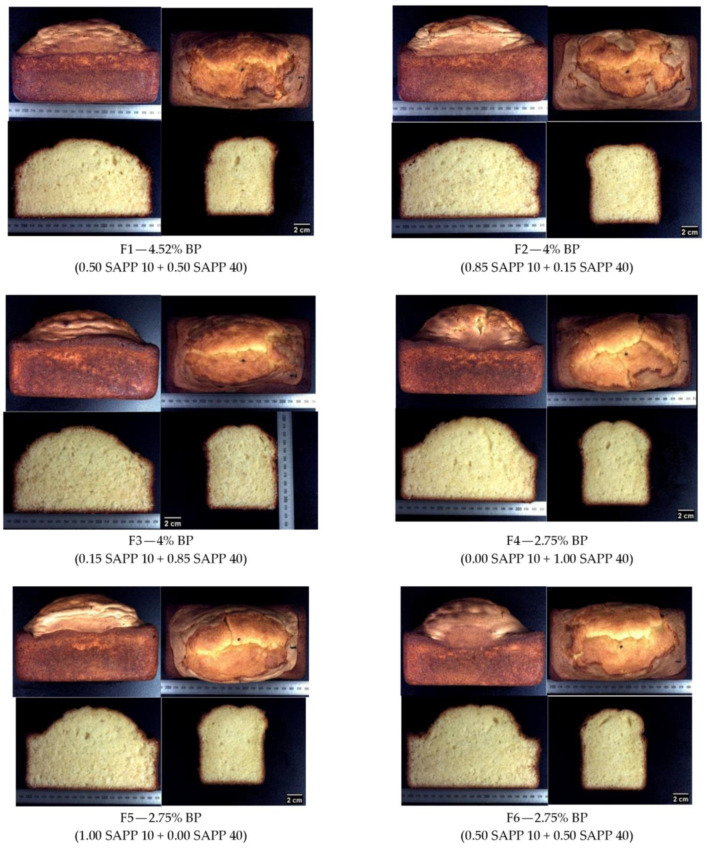
The appearance of pound cakes with different baking powder amounts and corresponding levels of SAPPs. The blue arrows indicate partially cooked parts of cake. F = formulation.

**Table 1 foods-12-00946-t001:** Pound cake batter recipe based on 100g of wheat flour.

Ingredients	Mass (g)
Wheat flour (T45)	100.00
Whole egg	84.75
Caster sugar	84.75
Sunflower oil	67.80
Refined salt	1.69
Baking powder	**X**

**X** = Different BP amounts with respective SAPP 10 and 40 levels presented in Table 2.

**Table 2 foods-12-00946-t002:** Generated batter and cake formulations from the central composite design of RSM/experimental design of BP and SAPP’s for batter and pound cake development.

Formulation	Baking Powder (%) = X	SAPP 10 (%) = a	SAPP 40 (%) = b
	[based on 100 g of WF proportion]	[based on the 58.1% of leavening acid component]	
1	4.52	50	50
2	4.00	85	15
3	4.00	15	85
4	2.75	0	100
5	2.75	100	0
6	2.75	50	50
7	2.75	50	50
8	1.50	85	15
9	1.50	15	85
10	0.98	50	50

The proportion of SAPP 40 is always the inverse of SAPP 10, but the sum of all SAPPs is equal to 100%.

**Table 3 foods-12-00946-t003:** Summary of significance level for the effect of BP, SAPP10 and SAPP40 on batter and cake properties.

	Parameter	Influence of (BP)	Influence of (SAPP10 and/or SAPP40)	Influence of (BP) and (SAPP10 and/or SAPP40)
Batter	Specific volume	+++	NS	+
	Porosity	+++	NS	+
	pH	NS	+++	NS
Cake	Specific volume	+++	+++	++
	Porosity	+++	+++	NS
	CO_2_ liberated	+++	+++	NS
	Total height	+++	NS	+++
	Base height	+++	+++	NS

NS—not significant, +—significant, ++—very significant, +++—extremely significant.

## Data Availability

Data are available upon request from the corresponding author.

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
