# Peer review of "Impact of Baking Powder and Leavening Acids on Batter and Pound Cake Properties"

_foods, 2023, doi:10.3390/foods12050946_

Round 1
Reviewer 1 Report
The authors investigated baking powder and leavening acids on batter and cake properties of pound cake. The topic is interesting, well-organized, and well-written, which could attract the readers of foods. However, some questions should be clarified. I recommend major revision.
Title
I suggest the following title of the manuscript: Impact of Baking Powder and Leavening Acids on Batter and Cake Properties of Pound Cake
Materials and Methods
Line 89, The protein content of wheat flour – type 45 is up to 11.5%. I mean that this wheat flour is not suitable as ingredient for making pound cake. Why do the authors choose it? Generally, the protein content of wheat flour should be less than 9% when it is used for making cake. Additionally, the authors should indicate the gluten content of wheat flour – type 45.
Line 90, What’s the meaning of wb? The authors should provide full name of the abbreviation appeared in the manuscript for the first time.
Line 94, baking powder, the content of sodium dicarbonate in baking powder should be reported. I think the content of sodium dicarbonate in baking powder is different when it is supplied by different producers. This causes the difference of the addition.
Line 99, Please provide the process of experiment design in the supplementary materials.
Line 101, baking powder is abbreviated as BP, leavening acid + leavening base is also BP. This confused me. Please differentiate.
Line 110, What’s the meaning of SB, please indicate.
Table 2, What’s the difference between formulation 6 and 7? Similarly, figure 5. Please check the central composite design. I am confused.
Line 158, for 45 minutes, I suggest 45 min.
Line 197, indicate the software used for experiment design, although the authors mentioned in section 2.2.
Line 199, “The Fisher’s…”, abundant space, please check.
Results and Discussion
Line 204, “surface methodology, (Table 2)”, please delete “,”
Table 3, CO2 liberated, please check.
Please report the results of batter specific volume, porosity, and pH, and cake specific volume, porosity, and CO2 liberated in the table. Additionally, please provide the regression equations for responses.
The result is lack of the verification experiment.
Conclusion
Line 349, Please check the format of this paragraph.
Reviewer 2 Report
Dear authors,
After reading the manuscript "Impact of Baking Powder and Leavening Acids Levels on Batter and Cake Properties of a Pound Cake ", I realized that the manuscript showed in some parts the scientific rigour wanted, but in other parts I have missed it.
The authors have presented critical evaluation only in some paragraphs.
The references are not current and it should be improved
Thats why, I have written some suggestions below in an attempt to improve the quality of the paper.
Title - L.2 - " Cake Properties of a Pound Cake " - I see no reason to have the word cake twice, besides as you have evaluated "Pound cake", using cake may give the idea that it would be the same behavior in all types of cake batter, which cannot be stated because it has not been evaluated.
L.17- " In this study, two leavening acids of sodium acid pyrophosphate (SAPP10 and SAPP40) were considered within the BP." The objetive that was presented in the abstract needs to be improved.
L.28- Avoid using the same words that are already in the title in the keywords, this can reduce the chance that other researchers find your paper, when they are searching articles.
L 33 and 34 - It got very repetitive in reading - "... leavening acid and a leavening base in the cake formulation. Chemical leavening... The same happened in the next paragraph, with the same word. Please, improve it as well.
L.49 - What is "[4] ?
L.78 - I missed in the Introduction why pound cake was chosen as a study option ? High consumption in your country ? in the world ? nothing was mentioned about the chosen product and the reader needs to be convinced of that as well.
L. 106- Give more details about salt (refined, pink salt, Himalayan salt ?)Why was caster sugar used ? Technological explanation ? Insert in the table that the eggs were liquid and that it was caster sugar.
L.140 - I couldn't understand why table 2 has treatment 6 and 7 and 1 and 10 only once.
L.150- pound cake - Figure 1
L.155 and 157 and so on - pound cake not cake
L.190 - Which is the author/ year of this methodology?
L.291- Please, standardize the fonts of all the figures, including the numerical scales, they look different sizes to me.
L.310 - I suggest including F9 - If I am not mistaken.
L.328- The "Appearance" attribute is not the only sensory characteristic that we can see, color can also be appreciated. I also suggest including statements about crust and crumb of the pound cake.
L- 290 - Were figures 3a, 3c and 4b quoted in the paper ? Check, please.
L.330- I suggest including F10. Please, check if all "F" were detailed.
L.331- instead of red circle, I would suggest blue arrows, since the picture is redish. And why only F10 got red circle. I really liked it.
L.339- 2.3.4."Conformation of cakes" - Shouldn't it also have been included in table 3 ?
L.339- " significance level on the effect of BP, SAPP10 and SAPP40 " - How was it determined? Author/Year ? Why is it not in the Material and methods that it would be performed? I also recommend discussing this result/table a bit, and not presenting the table before the conclusion of the paper.
L.350- "as reported in Table 3 on the other hand significantly" - The conclusion should answer your objectives, when you quote the table and mention being "significantly", it brings us back to the results.
Round 2
Reviewer 1 Report
Accept
Author Response
Thank you very much for accepting our responses, indeed, your comments and suggestions made our paper better.
Reviewer 2 Report
After another evaluation of the manuscript, I realized a great improvement in the quality of the paper.
The authors have accepted almost all of my requests, and those they did not accept, they have justified their point.
English is always useful to ask a native speaker for a final appreciation.
In relation to the Conclusion and table 3, the place is not adequate, I suggest to insert table 3 previously in Results and Discussion. I did not understand why it is in the middle of conclusion, at least for me.
Best wishes
Author Response
After another evaluation of the manuscript, I realized a great improvement in the quality of the paper.
The authors have accepted almost all of my requests, and those they did not accept, they have justified their point.
Thank you very much for your comments and suggestions, which has helped made the paper better.
English is always useful to ask a native speaker for a final appreciation.
Thank you very much for your advice, a thorough reading by a native speaker has been duly effected.
In relation to the Conclusion and table 3, the place is not adequate, I suggest to insert table 3 previously in Results and Discussion. I did not understand why it is in the middle of conclusion, at least for me.
Thank you for this remark. We probably misunderstood your previous comment about it when Table 3 was cited in the conclusion, which you pointed out that “it brings us back to the results”, this is why we moved it to the conclusion. In effect, the citation of Table 3 in the conclusion has been removed and the Table has been placed back in the Results and Discussion.